# KeSpeech: An Open Source Speech Dataset of Mandarin and Its Eight Subdialects

**Zhiyuan Tang[†], Dong Wang[‡], Yanguang Xu[†], Jianwei Sun[†], Xiaoning Lei[†]**
**Shuaijiang Zhao[†], Cheng Wen[†], Xingjun Tan[†], Chuandong Xie[†]**
**Shuran Zhou[†], Rui Yan[†], Chenjia Lv[†], Yang Han[†], Wei Zou[†], Xiangang Li[†]**
[†] Beike, Beijing, China
{tangzhiyuan001,zouwei026,lixiangang002}@ke.com
[‡] Tsinghua University, Beijing, China
wangdong99@mails.tsinghua.edu.cn

## Abstract

This paper introduces an open source speech dataset, KeSpeech, which involves 1,542 hours of speech signals recorded by 27,237 speakers in 34 cities in China, and the pronunciation includes standard Mandarin and its 8 subdialects. The new dataset possesses several properties. Firstly, the dataset provides multiple labels including content transcription, speaker identity and subdialect, hence supporting a variety of speech processing tasks, such as speech recognition, speaker recognition, and subdialect identification, as well as other advanced techniques like multi-task learning and conditional learning. Secondly, some of the text samples were parallel recorded with both the standard Mandarin and a particular subdialect, allowing for new applications such as subdialect style conversion. Thirdly, the number of speakers is much larger than other open-source datasets, making it suitable for tasks that require training data from vast speakers. Finally, the speech signals were recorded in two phases, which opens the opportunity for the study of the time variance property of human speech. We present the design principle of the KeSpeech dataset and four baseline systems based on the new data resource: speech recognition, speaker verification, subdialect identification and voice conversion. The dataset is free for all academic usage.

## 1 Introduction

Deep learning empowers many speech applications such as automatic speech recognition (ASR) and speaker recognition (SRE) [1, 2]. Labeled speech data plays a significant role in the supervised learning of deep learning models and determines the generalization and applicability of those models. With the rapid development of deep learning for speech processing, especially the end-to-end paradigm, the size of supervised data that is required grows exponentially in order to train a feasible neural network model. A lot of open source datasets have been published to meet this urgent need, and many of them are available on OpenSLR[1]. For example, LibriSpeech [3] is a very popular open source English ASR dataset with a large-scale duration of 1,000 hours, and Thchs-30 [4] and AISHELL-1/2 [5, 6] are usually used for the training and evaluation of Chinese ASR. Recently, very large-scale open source speech corpora emerge to promote industry-level research, such as the GigaSpeech corpus [7] which contains 10,000 hours of transcribed English audio, and The People's Speech [8] which is a 31,400-hour and growing supervised conversational English dataset. In the scope of SRE, less task-specific datasets have been open-sourced as many ASR corpora with speaker identity can be repurposed. The most commonly referenced SRE corpora are VoxCeleb1/2 [9, 10]

---

[1]http://openslr.org

and CN-Celeb [11] collected from public media and spoken with English (mostly) and Chinese respectively.

Although the increasing amount of open source supervised speech data facilitates the high data-consuming deep learning training, there still remains many challenges in terms of data diversity and scale. Precisely, most speech datasets were designed with single label for a specific task, preventing effective multi-task learning and conditional learning on other information, such as the joint learning of ASR and SRE, phonetic-aware SRE and language-agnostic SRE. Moreover, many of them focus on most spoken languages with standard pronunciation, such as American English and standard Chinese, regardless of dialect or heavy accent, that hurts the diversity of language research and the protection of minority languages or dialects. As for Chinese ASR, due to the rich variety of Chinese dialects and subdialects, the appeal to dialect speech corpus is much more urgent. As for SRE without consideration of language dependency, limited number of speakers in existing open source corpora prevents deep learning from taking a step further, considering VoxCeleb only contains not more than 8,000 speakers.

In this paper, we introduce a large scale speech dataset named *KeSpeech* which was crowdsourced from tens of thousands of bro**ke**rs with the Bei**ke** platform for real estate transactions and housing services in China. Specifically, the KeSpeech dataset contains 1,542 hours of transcribed audio in Mandarin (standard Chinese) or Mandarin subdialect from 27,237 speakers in 34 cities across China, making it possible for multi-subdialect ASR and large-scale SRE development. Besides the labels of transcription and speaker information, KeSpeech also provides subdialect type for each utterance, which can be utilized for subdialect identification to determine where the speaker is from. Overall, the KeSpeech dataset has special attributes as follows:

- Multi-label: The dataset provides multiple labels like transcription, speaker information and subdialect type to support a variety of speech tasks, such as ASR, SRE, subdialect identification and their multi-task learning and conditional learning on each other. How subdialect and phonetic information influence the performance of SRE model can also be investigated.

- Multi-scenario: The audio was recorded with mobile phones from many different brands in plenty of different environments. The massive number of speakers, both female and male, are from 34 cities with a wide range of age.

- Parallel: Part of the text samples were parallel recorded in both standard Chinese and a Mandarin subdialect by a same person, allowing for easily evaluating subdialect style conversion and more effective subdialect adaptation in Chinese ASR. To what extent subdialect could affect the SRE can also be investigated.

- Large-scale speakers: As far as we know, the speaker number from this dataset reaches the highest among any other open source SRE corpora, that can obviously boost the industry-level SRE development.

- Time interval: There is at least a 2-week interval between the two phases of recording for most of the speakers (17,558 out of total 27,237), allowing for the study of time variance in speaker recognition. The speakers probably had different background environments for the second recording, resulting in more diverse data which is favorable for generalization in ASR training.

After the description of the dataset and its construction, we conduct a series of experiments on several speech tasks, i.e., speech recognition, speaker verification, subdialect identification and voice conversion, to demonstrate the value the dataset can provide for those tasks and inspirations by analyzing the experiment results. We also do some discussion about the limitations, potential risks and research topics related to this dataset.

## 2 Related Work

KeSpeech supports multiple speech tasks, while during the review of related work, we focus on ASR and SRE which the dataset can benefit the most. KeSpeech focuses on Chinese, so we present some open source benchmark speech corpora for Chinese related research and development, and compare them with KeSpeech in terms of duration, number of speakers and label in Table 1. As most SRE

Table 1: Comparison between benchmark speech datasets and KeSpeech (In the literature, Mandarin generally stands for standard Chinese, though it actually indicates a group of subdialects).

| Dataset | Language | #Hours | #Speakers | Label |
|---|---|---|---|---|
| Thchs-30 [4] | Mandarin | 30 | 40 | Text, Speaker |
| AISHELL-1 [5] | Mandarin | 170 | 400 | Text, Speaker |
| AISHELL-2 [6] | Mandarin | 1,000 | 1,991 | Text, Speaker |
| HKUST/MTS [12] | Mandarin | 200 | 2,100 | Text, Speaker |
| DiDiSpeech [13] | Mandarin | 800 | 6,000 | Text, Speaker |
| VoxCeleb1/2 [9, 10] | English Mostly | 2,794 | 7,363 | Speaker |
| CN-Celeb [11] | Mandarin | 274 | 1,000 | Speaker |
| KeSpeech | Mandarin and 8 Subdialects | 1,542 | 27,237 | Text, Speaker, Subdialect |

practice is independent of language, we present several open source SRE benchmark speech corpora regardless of language, dialect or accent. The comparison between SRE benchmark datasets and ours is also presented in Table 1.

Table 1 shows that the previous open source Chinese ASR corpora were generally designed for standard Chinese or accented one from limited region, regardless of multiple dialects and wide geographical distribution. On the other hand, commonly used SRE benchmark datasets have relatively small number of speakers, and their lack of text transcription makes it hard to do phonetic-aware SRE training or joint learning with ASR. KeSpeech is expected to diminish those disadvantages by providing multiple subdialects and large scale speakers across dozens of cities in China, and is also highly complementary to existed open source datasets.

## 3 Data Description

### 3.1 License

The dataset can be freely downloaded[2] and noncommercially used with a custom license[3]. Besides the preset tasks in the dataset directory, the users can define their own ones under the license.

### 3.2 Mandarin and Its Eight Subdialects

China has 56 ethnic groups with a quite wide geographical span, thus resulting in a great variety of Chinese dialects. Besides the official standard Chinese, the active Chinese dialects can be generally divided into ten categories, i.e., Mandarin dialect, Jin dialect, Wu dialect, Hui dialect, Min dialect, Cantonese, Hakka, Gan dialect, Xiang dialect and Pinghua dialect, among which Mandarin is the most spoken one by more than 70% of the Chinese population and is also the basis of standard Chinese [14].

Standard Chinese originated from Northern Mandarin, so it can also be described as standard Northern Mandarin, standard Beijing Mandarin, standard Mandarin or simply Mandarin [15]. In the literature of most Chinese speech corpora and ASR research, when speaking of Mandarin, it often refers to standard Chinese. In this paper and KeSpeech, the single word 'Mandarin' stands for standard Chinese or some accents very close to standard Chinese. Nevertheless, as a group of dialects, Mandarin still contains many distinguishable subdialects far from standard Chinese. According to the predominant number of people who speak it, the Language Atlas of China [14] divides Mandarin into eight subdialects which are the focus of KeSpeech. The eight Mandarin subdialects and their locations of city or province are described as follows (some cities or provinces may be involved in several subdialects):

- Beijing Mandarin: Though named after the city Beijing, this subdialect is composed of a number of similar dialects spoken by many areas in North China including Beijing, Tianjin, Hebei, Chaoyang and Chifeng. Specially, Beijing Mandarin lays the foundation for standard Chinese and Luanping County, the home of standard Chinese in Hebei, provides the phonological specification for standard Chinese.

[2]https://github.com/KeSpeech/KeSpeech
[3]https://github.com/KeSpeech/KeSpeech/blob/main/dataset_license.md

- Southwestern Mandarin or Upper Yangtze Mandarin: This subdialect is spoken in Southwest China including Sichuan, Chongqing, Yunnan, Guizhou, Hubei, Hunan and Guangxi.
- Zhongyuan Mandarin or Central Plains Mandarin: This subdialect is spoken in Central China including Shaanxi, Henan, Shanxi, Gansu, Hebei, Anhui, Jiangsu, Xinjiang and Shandong.
- Northeastern Mandarin: This subdialect is spoken in Northeast China including Heilongjiang, Jilin, Liaoning, Hebei and Inner Mongolia.
- Lan-Yin Mandarin: The name is a compound of two capitals of two adjacent provinces, i.e. Lanzhou of Gansu and Yinchuan of Ningxia, so this subdialect is mainly spoken by people from Lanzhou and Yinchuan and their nearby areas even including some parts in Xinjiang and Inner Mongolia.
- Jiang-Huai Mandarin or Lower Yangtze Mandarin: Jiang-Huai means the area around Changjiang (Yangtze river) and Huai river, including part of Anhui, Hubei, Jiangsu and Jiangxi, so this subdialect encompasses similar dialects spoken in those places.
- Ji-Lu Mandarin: Ji and Lu are the abbreviations of two provinces Hebei and Shandong respectively, so this subdialect involves similar dialects in most areas of the two provinces and nearby areas in Tianjin, Beijing, Shaanxi and Inner Mongolia.
- Jiao-Liao Mandarin: Jiao and Liao stand for two peninsulas, i.e., Jiaodong and Liaodong, which are connected by the Bohai Sea. The representative cities in these two peninsulas include Dalian, Dandong, Yingkou, Yantai, Weihai and Qingdao. This subdialect is also spoken in some areas in Jilin next to Liaodong.

As for mutual intelligibility with standard Chinese in terms of both pronunciation and text, Northern subdialects, i.e., Beijing and Northeastern, are obviously on the top, and other Mandarin varieties can also be roughly understood by people who speak standard Chinese. Each subdialect has its own special vocabulary, while vocabulary of standard Chinese constitutes the main part of all subdialects. That is to say, any sentence from standard Chinese can be spoken by Mandarin subdialects.

### 3.3 Dataset Structure and Label

The dataset is published as a data directory, named KeSpeech, which contains three subdirectories, i.e., Audio, Metadata and Tasks. The Audio directory contains all the audio files with WAV format recorded with 16 kHz sampling rate. The Tasks directory presets several speech tasks based on the whole dataset including speech recognition, speaker verification and subdialect identification, which will be elaborately described in Section 4. The Metadata presents all label information and other related things with Kaldi style [16], and the details are described as follows:

- city2Chinese, subdialect2Chinese, city2subdialect: The first two files shows the relationship between English name and Chinese characters for both city and subdialect. The city2subdialect indicates which subdialect is mainly spoken in the city.
- phase1.text, phase1.wav.scp, phase2.text, phase2.wav.scp: These files tell the transcription text and audio path for each utterance for data collection phase 1 and 2. Each utterance has a unique identity number. There is an interval of two weeks between phase 1 and 2.
- phase1.utt2style, phase1.utt2subdialect: As in the phase 1 of data collection, we have samples spoken with Mandarin or its subdialects, the former file tells whether an utterance was labeled by human as 'Mandarin' or 'Dialect'. With phase1.utt2subdialect, one can find which subdialect each utterance is exactly spoken with.
- phase2.utt2env: All data collected in phase 2 was spoken with Mandarin, while we provide the background environment information labeled by human as 'clean' or 'noisy'.
- spk2age, spk2city, spk2gender, spk2utt, utt2spk: These files describe the information of the speakers, such as their age, gender, city where they were born and what utterances they have spoken. Each speaker has a unique identity number.

### 3.4 Statistics

There are a total of 27,237 speakers in the dataset with 9,739 females and 17,498 males. Figure 1 shows the speaker distribution with respect to age and gender. Males who are younger are almost

twice as many as the females in the dataset. The number of females starts to catch up as the age increases. Figure 2 shows the speaker distribution in 34 cities involved in KeSpeech. The quantity imbalance occurs due to a large difference in number of volunteers from different cities. Table 2 shows how many hours, utterances and speakers each subdialect contains. This will further instruct us to prepare training set and test set for different speech tasks. In addition, some speakers spoke both Mandarin and subdialect, and phase 1 and 2 of data collection share most of all speakers.

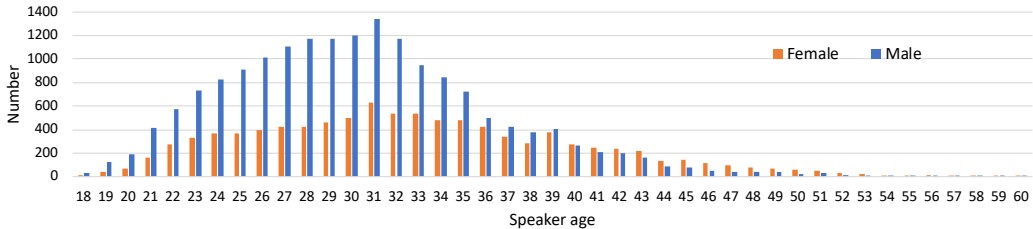

Figure 1: Number of speakers with different ages for female and male in KeSpeech.

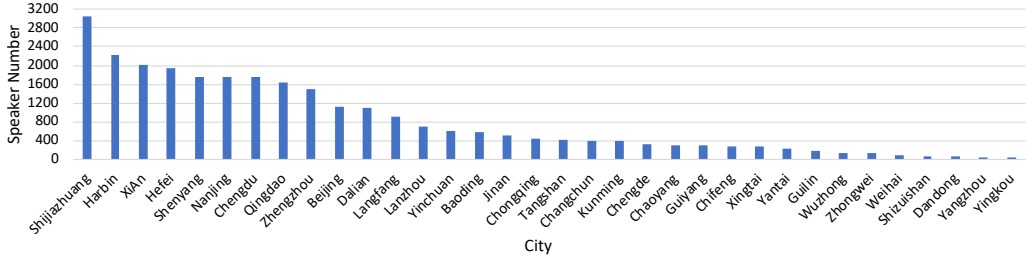

Figure 2: Number of speakers from 34 cities in KeSpeech.

Table 2: Number of hours, utterances and speakers for Mandarin and each subdialect for two phases of data collection in KeSpeech (As mentioned above, single Mandarin generally stands for standard Chinese).

| Phase | Subdialect | #Hours | #Utterances | #Speakers |
|---|---|---|---|---|
| 1 | Mandarin | 589 | 370,819 | 22,621 |
| | Zhongyuan Mandarin | 84 | 52,012 | 3,038 |
| | Southwestern Mandarin | 75 | 48,465 | 2,646 |
| | Ji-Lu Mandarin | 59 | 36,921 | 2,832 |
| | Jiang-Huai Mandarin | 46 | 30,008 | 2,913 |
| | Lan-Yin Mandarin | 35 | 22,324 | 1,215 |
| | Jiao-Liao Mandarin | 34 | 21,847 | 1,968 |
| | Northeastern Mandarin | 8 | 5,205 | 1,264 |
| | Beijing Mandarin | 4 | 2,538 | 386 |
| 2 | Mandarin | 608 | 385,915 | 17,750 |
| Totally | | 1,542 | 976,054 | 27,237 |

## 3.5 Data Construction

### 3.5.1 Crowdsourcing

**Text Selection** We randomly selected 1.4 million text segments from the news genre in CLUECorpus2020-small [17] as the text source for recording, and each candidate segment generally contained 12 to 18 Chinese characters. The news genre in CLUECorpus2020-small was crawled from the public We Media (self-media) platform and it involves almost all common topics like sports, finance, entertainment and technology. Finally, KeSpeech used 729,972 unique text segments.

**City and Broker Selection** We recruited volunteers for recording audio from brokers working on the Beike platform, while we needed firstly to determine which broker the recording task should be delivered to. We suppose that the broker may probably speak the local dialect where he or she

was born or even grown. With the Beike platform we could find each broker's city of birth, so we selected several relatively big or representative cities for each subdialect, and then pushed the task to brokers in those cities with a request of recording with their familiar dialect. The cities selected for each subdialect are listed as follows:

- Beijing Mandarin: Beijing, Chaoyang, Chengde, Chifeng, Langfang

- Jiang-Huai Mandarin: Hefei, Nanjing, Yangzhou

- Jiao-Liao Mandarin: Dalian, Dandong, Qingdao, Weihai, Yantai, Yingkou

- Ji-Lu Mandarin: Baoding, Jinan, Shijiazhuang, Tangshan, Xingtai

- Lan-Yin Mandarin: Lanzhou, Shizuishan, Wuzhong, Yinchuan, Zhongwei

- Northeastern Mandarin: Changchun, Harbin, Shenyang

- Southwestern Mandarin: Chengdu, Chongqing, Guilin, Guiyang, Kunming

- Zhongyuan Mandarin: Xi'an (XiAn), Zhengzhou

**Task Performing** After the preparation of candidate text pool for recording audio and targeted brokers who might speak a dialect in their birthplace, we delivered the recording task to the brokers through a mobile application which was used by all registered brokers on the platform. The application supports different kinds of crowdsourcing tasks including audio recording by reading some text and it records audio with WAV format by 16 kHz sampling rate. The task required each speaker to produce at least 20 utterances whose text was randomly picked from the text pool. The participants were rewarded with a simple virtual currency which could only be used on the platform. We finished the crowdsourcing task when the growth of audio quantity became very slow as less people participated in the task, then we collected all data for postprocessing.

**Parallel Task** We built another crowdsourcing task, i.e. phase 2, with an interval of two weeks after the above task, i.e. phase 1. The phase 2 task aimed to collect standard Mandarin audio samples parallel to subdialect ones in phase 1 in terms of same speaker and same text. The crowdsourcing procedure of phase 2 was almost the same as phase 1 except that the targeted speakers and text were already used in phase 1. After combination of the data from phase 1 and 2, we found that many utterances in phase 1 were not spoken in subdialect but Mandarin, and some speakers just participated in only one of the two phases.

Notably, candidate brokers had to sign an agreement [4] for open-sourcing their speech for academic usage before further recording for the two crowdsourcing tasks.

### 3.5.2 Postprocessing

**Transcription Validation and Subdialect Labeling** All audio data was collected after completion of the above two crowdsourcing tasks. We then employed two professional data companies to do the manual labeling with three steps, i.e., labeling by one person, full inspection by another person and random inspection by a third person. To increase the labeling quality, we split the audio samples per person of each phase to two parts to the two data companies respectively. For data from phase 1, the audio was labeled with 'Mandarin' or 'Dialect' by a native person from the city where its speaker was born. The label Mandarin means the audio is almost standard Mandarin, and Dialect means the auditor thought it's the local dialect. For data from phase 2, the audio was not additionally labeled by dialect as it's all Mandarin, while it was labeled 'clean' or 'noisy' by the auditor according to its background condition. For data from both phase 1 and 2, the transcription of each audio was checked and corrected, as some of the audio did not actually correspond to the reference text.

**Anonymization** The speaker identity information was anonymized so as to remove any relationship with the volunteers on the Beike platform. Other non-sensitive speaker information such as age and gender was acquired from the registration on the platform.

---

[4]https://github.com/KeSpeech/KeSpeech/blob/main/volunteer_agreement.md

# 4  Tasks and Baselines

KeSpeech presets several speech tasks, i.e. speech recognition, speaker verification and subdialect identification, by preparing training set, development set and test set, which serve for benchmark training and evaluation. Voice conversion is also conducted to demonstrate potential usage.

## 4.1  Speech Recognition

Each set of ASR task has a file 'utt2subdialect' indicating the subdialect an utterance belongs to, with which we can regroup the set into ones containing only one type of subdialect and then recombine them as needed. Specifically, the development and test sets contain randomly selected 100 and 2,000 speakers with all their audio samples respectively, and the speakers were involved in both phase 1 and 2. We build different baselines according to the selection of training data from the whole training set. All baselines follow the pipeline and configuration provided by a state-of-the-art ESPnet ASR recipe[5], and are trained with 8 GPUs of NVIDIA V100.

Table 3: CER(%) results of ASR baselines (BJ: Beijing, ZY: Zhongyuan, SW: Southwestern, NE: Northeastern, LY: Lan-Yin, JH: Jiang-Huai).

| Training subset | Test subset | | | | | | | | | AISHELL-1 | |
| --- | --- | --- | --- | --- | --- | --- | --- | --- | --- | --- | --- |
| | Mandarin | BJ | SW | ZY | NE | LY | JH | Ji-Lu | Jiao-Liao | dev | test |
| Each subdialect | - | 97.4 | 18.3 | 16.9 | 54.4 | 25.4 | 29.2 | 22.3 | 26.0 | - | - |
| All subdialects | 13.7 | 17.5 | 18.1 | 16.0 | 18.9 | 16.7 | 22.6 | 16.4 | 17.4 | 11.9 | 11.9 |
| Mandarin | 6.9 | 17.9 | 17.7 | 18.7 | 12.1 | 22.1 | 20.6 | 15.8 | 15.5 | 5.7 | 6.0 |
| Whole training set | 6.1 | 11.5 | 11.9 | 9.6 | 10.2 | 11.7 | 15.9 | 11.5 | 11.7 | 5.2 | 5.3 |

Table 3 presents the Character Error Rate (CER) results for each subdialect and Mandarin subset in the test set respectively with different selection of training data. In the first line of the results, each subdialect in test set is evaluated on its own ASR system trained only on that subdialect from the whole training set, while the second line shares the same model trained on all combined subdialects, the third line shares the same model trained on all Mandarin, and the last line has its model trained on the whole training set. In the results, the high CERs for Beijing and Northeastern Mandarin with their own ASR models are due to the scarcity of training data as shown in Table 2, which can be greatly decreased when other subdialects or Mandarin are involved for training.

Obviously, Mandarin training data can benefit the subdialects, especially similar ones, and vice versa, while there is still much work to do on subdialect recognition to catch up with the performance on Mandarin. With the model trained with all subdialects, the performance on Mandarin is better than subdialects, indicating that Mandarin shares much with all subdialects. Figure 3 presents CER matrix for 3 best trained subdialect ASR models, indicating the difference along with similarity in the subdialects. The last 3 baseline models are also evaluated on the development and test sets from the commonly used Mandarin dataset AISHELL-1 [5] as shown in the last two columns of Table 2 to further verify the reasonability of dataset construction and model building.

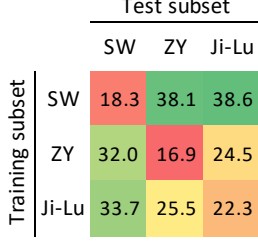

Figure 3: CER(%) matrix of 3 subdialects.

## 4.2  Speaker Verification

Different with other tasks, speaker verification has a development set for training and four evaluation sets for analysis on different variations, including time interval and subdialect type. The evaluation sets reorganized from the test set of above ASR are described in Table 4. Three SRE models are trained respectively with 2 GPUs of NVIDIA V100 following the pipeline and configuration of popular open-source SRE recipes, i.e., Kaldi-xvector[6] without data augmentation, ResNet-34[7] and ECAPA-TDNN[8].

[5]https://github.com/espnet/espnet/tree/master/egs/aishell/asr1
[6]https://github.com/kaldi-asr/kaldi/blob/master/egs/sre16/v2
[7]https://github.com/Snowdar/asv-subtools/blob/master/recipe/voxcelebSRC
[8]https://github.com/Snowdar/asv-subtools/blob/master/pytorch/model/ecapa-tdnn-xvector.py

Table 4: Evaluation sets for speaker verification (Enrollment and Test have exactly the same speakers).

| Evaluation set | Enrollment | Test |
|---|---|---|
| Eval-combined | Utterances from both phase 1 and 2 | Utterances from both phase 1 and 2 |
| Eval-phase1-phase2 | Utterances from phase 1 | Utterances from phase 2 |
| Eval-Mandarin-Mandarin | Mandarin utterances from phase 1 | Mandarin utterances from phase 2 |
| Eval-subdialect-Mandarin | Subdialect utterances from phase 1 | Mandarin utterances from phase 2 |

Table 5: Results of four evaluation sets for speaker verification.

| Model | Eval-combined | | Eval-phase1-phase2 | | Eval-Mandarin-Mandarin | | Eval-subdialect-Mandarin | |
|---|---|---|---|---|---|---|---|---|
| | EER(%) | MinDCF | EER(%) | MinDCF | EER(%) | MinDCF | EER(%) | MinDCF |
| Kaldi-xvector | 0.47 | 0.0455 | 5.46 | 0.4010 | 4.59 | 0.4406 | 8.3 | 0.4982 |
| ResNet-34 | 0.41 | 0.0319 | 4.57 | 0.3626 | 4.55 | 0.3669 | 5.8 | 0.5384 |
| ECAPA-TDNN | 0.32 | 0.0255 | 3.86 | 0.4547 | 3.78 | 0.4504 | 4.8 | 0.4425 |

Table 5 reports results on the evaluation sets in terms of equal error rate (EER) and the minimum of the normalized detection cost function (DCF) at $P_{Target} = 10^{-2}$. The combined evaluation achieves significantly better performance among others due to data diversity of enrollment and test set. The worse performance of both Eval-phase1-phase2 and Eval-Mandarin-Mandarin indicates that time variance has a negative impact on SRE. The gap between Eval-Mandarin-Mandarin and Eval-subdialect-Mandarin can be attributed to subdialect switch between enrollment and test.

### 4.3 Subdialect Identification

We reuse the training, development and test sets of above ASR for this task but just keeping the data of phase 1, and each set has a file 'utt2subdialect' indicating the subdialect an utterance belongs to, which serves as the supervision this task needed. We train three models by adopting above SRE architectures and following the same pipeline and configuration except the label preparation. As the data of Northeastern and Beijing Mandarin is not enough for training a reasonable deep learning system as shown in Table 2 and 3, we just use the rest 6 subdialects for experiment.

Table 6: Results of subdialect identification (JH: Jiang-Huai, LY: Lan-Yin, SE: Southeastern, ZY: Zhongyuan).

| Model | Accuracy(%) per subdialect | | | | | | Overall | |
|---|---|---|---|---|---|---|---|---|
| | Ji-Lu | JH | Jiao-Liao | LY | SE | ZY | Accuracy(%) | $C_{avg}$ |
| Kaldi-xvector | 43.22 | 53.19 | 58.56 | 43.76 | 82.47 | 52.63 | 56.34 | 0.2376 |
| ResNet-34 | 39.37 | 64.44 | 37.84 | 42.67 | 82.32 | 76.65 | 61.13 | 0.2095 |
| ECAPA-TDNN | 48.45 | 68.56 | 36.52 | 44.18 | 80.91 | 68.54 | 60.77 | 0.2147 |

Table 6 reports identification accuracy of the three models on each subdialect, and the overall accuracy and $C_{avg}$ metric described in NIST Language Recognition Evaluation [18]. The results for most subdialects are quite unsatisfying, that may be attributed to the similarity between some subdialects. Figure 4 shows the t-SNE [19] result on ResNet-34 embeddings of the subdialects with randomly selected 300 utterances per one, which is consistent with the accuracy that most of the subdialects are mixed up and hard to distinguish.

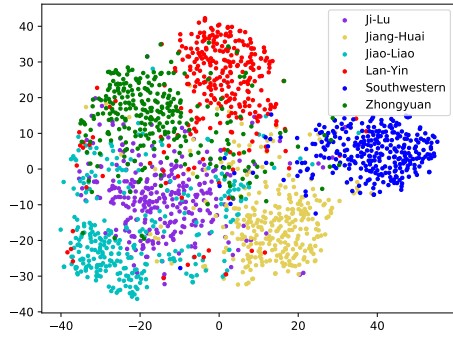

Figure 4: t-SNE result of 6 subdialects.

### 4.4 Voice Conversion

To demonstrate potential usage of the dataset, we also conduct a toy cross-subdialect voice conversion experiment with the Fastspeech-VC framework [20], that is, enveloping a source utterance in one subdialect with the timbre of target person who speaks another subdialect. This task is not preset in KeSpeech.

In the evaluation, source utterances are selected from one male (speaker A) and female (speaker B) who can speak Zhongyuan and Jiao-Liao Mandarin respectively. Besides subdialect utterances, the source utterances also contain some Mandarin (standard Chinese) from the same speakers A and B. Target speaker information is conveyed by some Southwestern Mandarin utterances. All audio samples in this section are available at our demo page[9].

Table 7: MOS on naturalness of converted voices and speaker similarity between converted voices and those of target speaker.

| Converted voice | Naturalness | Speaker similarity |
|---|---|---|
| Ground truth of target speaker | $4.61 \pm 0.10$ | 1.000 |
| Zhongyuan Mandarin (From speaker A) | $4.07 \pm 0.08$ | 0.892 |
| Mandarin (From speaker A) | $4.16 \pm 0.08$ | 0.890 |
| Jiao-Liao Mandarin (From speaker A) | $4.24 \pm 0.09$ | 0.878 |
| Mandarin (From speaker B) | $4.36 \pm 0.08$ | 0.872 |

Table 7 presents mean opinion score (MOS) of naturalness from 10 female and 10 male Chinese by scoring with bad (1), poor (2), fair (3), good (4) or excellent (5). The table also presents speaker similarity between converted voices and those of target speaker by computing the cosine similarity between the speaker embeddings extracted with above SRE Kaldi-xvector. The results reveal that the voice can be converted independent of subdialect, and more complex conversion can be explored.

## 5 Discussion

### 5.1 Data Limitations

**Subjectivity of Labeling** Although the labeling procedure was conducted by professional data companies, the labeling of subdialect type was of much subjectivity for some accented utterances that can not be classified definitely into Mandarin (standard Chinese) or subdialect.

**Data Imbalance** Due to the broker distribution on the platform, speaker distribution is not very balanced in each subdialect or city, so as the audio duration. All subdialects account for only a third of the total duration and two subdialects have not enough training data to build a feasible ASR system from scratch.

**Diversity inside Subdialect** Each subdialect often covers a large area in China, resulting in many similar dialects inside that subdialect, that hurts subdialect identification. Furthermore, some subdialects, such as those in North China, are too similar to be distinguished.

### 5.2 Potential Risks

As the text was from open-source corpus with no private or sensitive information, the biggest ethical concern in this dataset is about speaker identity. Although all speaker identity numbers were randomly generated and have no relationship at all with the original ones, there still exists some potential risks that one can de-anonymize that by, for example, comparing the whole dataset to the voice of a specific broker on the platform to retrieve all his/her audio for malicious purposes. On the other hand, all speakers comprise a subset of the Beike brokers, so the platform (if voice authentication is allowed) may be invaded with voice of any one regardless of the specific identity. To reduce

---

[9]https://wen2cheng.github.io

the risks as much as possible, we host and maintain the dataset on Beike's own cloud for better monitoring data usage. With the dataset's license, any academic institutions should detail their research plan with a request form, and we will evaluate it carefully. We will also inspect if the data is validly used from time to time, by following the work of users or communicating with them directly.

## 5.3 Research Topics

**Robust ASR and SRE** The multi-subdialect and multi-scenario information in KeSpeech can be employed to build robust ASR and SRE. Effective subdialect adaptation for existed ASR models and transfer learning between Mandarin and subdialect can also be studied. The subdialect type and time variance all have bad influence on usual SRE, that appeals for robust SRE training with phonetic-aware, subdialect-agnostic or time-invariant methods.

**Speech Factorization** With multiple labels in KeSpeech, the speech can be disentangled into factors including at least subdialect type, speaker identity and content, with which subdialect conversion and other reconstruction can be explored.

**Multi-task Learning** Different speech tasks can benefit each other by giving complementary information or counteracting task-independent information, while performing all tasks in one model like human is full of challenges. KeSpeech, combined with other corpora, will promote this direction.

## 6 Conclusions

This paper introduced a large-scale open source speech corpus, which was crowdscourced from tens of thousands of volunteers and recorded with the most spoken dialect Mandarin in China and its eight subdialects. The dataset is structurally organized and presets several benchmark speech tasks for academic reference, i.e., speech recognition, speaker verification and subdialect identification whose baseline results were also presented. The paper also described the limitations, potential risks and research directions for the dataset.

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
