# OpenReview forum: "KeSpeech: An Open Source Speech Dataset of Mandarin and Its Eight Subdialects"
_NeurIPS.cc/2021/Track/Datasets_and_Benchmarks/Round2 — NeurIPS 2021 Datasets and Benchmarks Track (Round 2)_

### Official Review · Reviewer_91VF · 2021-09-11
**Great dataset for subdialects with wide speaker variety**

**Rating:** 7
**Confidence:** 3
**Clarity:** Mostly.

**Strengths:**

1. Dataset offers a large number of speakers from diverse locations in China, about 4X more than existing datasets

2. Dataset offers labels for subdialect. Dealing with dialects is a significant issue and generally doesn't get enough attention. This is very  cool!

3. Dataset offers diverse context, e.g., different recording devices, noisy background noise, etc.

4. Authors performed initial analysis of gender split and speaker location

5. The experiments especially table 3 illustrates the importance of subdialects and the challenge, really nice!

**Weaknesses:**

1. The license is problematic: CC-BY-NC-SA. The non-commercial clause will limit usage by industry researchers.

2. The labels are crowd sourced and therefore of unknown quality, I would like to see this explored more thoroughly.

The authors write: "Each audio was then listened by people to check and correct its transcription, as some of the audio did not actually correspond to the reference text."

Could they elaborate and help me understand this better? I think it's possible this isn't a problem, but need more info. Who listened to check the crowd-sourced work? Were the listeners skilled?

3. The transcription and audio are probably not aligned, which will generally impede use for ASR.

4. I did not see an analysis of how much of the data was noisy versus clean. Also it looks like noisy vs. clean was only labeled on Mandarin.

5. I don't see a hosting/maintenance plan.

6. No noise vs. clean on dialects

If items 2 and 5 are fixed, it would change my rating.

**Additional Feedback:**

You should probably cite: https://openreview.net/forum?id=R8CwidgJ0yT

Generally this is really cool work and I'd like for it to get published. Things your team could do that would change my rating  and help push it up above publishing threshold:

1. Align the audio and transcription to enable ASR (or convince me it's not necessary)
2. More detail on label quality to help future users
3. Hosting and maintenance plan
4. Noisy vs. clean on dialects (I know this is a big lift and may not be possible)
5. Use a CC-BY license

This has the possibility of being a 6-8 with some additional work.

I've upgraded to 7.

**Correctness:**

The audio and transcripts should be aligned.

I would be happier if we had noise labels on the dialects instead of just Mandarin.

**Documentation:**

Data collection and organization look  good.

Ethics are probably ok, I'm not an expert.

I don't have documentation. The label descriptions need improvement as per above.

I don't see a hosting and maintenance plan.

**Ethics:**

Seems ok, I'm not an expert.

**Relation To Prior Work:**

Yup.

**Summary And Contributions:**

Good dataset that will improve our ability to understand dialects of Chinese. Generally this area is underexplored and I'm pretty excited. I think this paper could be a 6-8 with additional work.

10/1/2021 update: Great work addressing my feedback, based on the improvements in the paper and underlying work I would upgrade my  rating to a 7!

Congratulations to the authors, look forward to seeing the data set.

---

> ### Author Response · Authors · 2021-09-26
> **Reply to Reviewer 91VF**
>
> Thanks a lot for your careful review and valuable suggestions.
> We updated the paper and reply to the concerns one by one.
>
>
> * Non-commercial license.
>   - As the dataset was supported by a listed company, the company legal dept suggested a non-commercial license.
>   - Participants signed an agreement for open-sourcing their speech for academic usage. If the agreement allows for commercial usage, it may reduce the enthusiasm of volunteers.
>   - We used the text from CLUECorpus2020-small which is non-commercial.
>
> * Labeling procedure.
>   - We revised the paper to elaborate the procedure. **(line #221)**
>   - The procedure for quick reference:
>     - 1. We employed two professional data companies to do the manual labeling with three steps, i.e., labeling by one person, full inspection by another person and random inspection by a third person. Those people were trained.
>     - 2. To increase the labeling quality, we split the audio samples per person of each phase to two parts to the two data companies respectively.
>     - 3. For data from phase 1, the audio was labeled with *Mandarin* or *Dialect* by a **native person** from the city where its speaker was born.
>     - 4. For data from phase 2, the audio was not additionally labeled by dialect as it's all Mandarin, while it was labeled *clean* or *noisy* by the auditor according to its background condition.
>     - 5. For data from both phase 1 and 2, the transcription of each audio was checked and corrected, as some of the audio did not actually correspond to the reference text.
>
> * Align the audio and transcription.
>   - 1. The audio was recorded by reading reference text, so most of them are well aligned.
>   - 2. As described in labeling procedure, we employed two professional data companies to do the manual labeling, including checking and correcting the transcription.
>   - 3. We evaluated our models on common-used AISHELL-1 dev/test, the results verified that the dataset is reasonable for training ASR. **(Table 3)** We paste the CER% results below for quick reference.
>
>      | Model |  AISHELL-1 dev | AISHELL-1 test |
>      |--|--|--|
>      |All subdialects (ours) | 11.9| 11.9  |
>      |Mandarin (ours)        | 5.7 | 6.0 |
>      |Whole training set (ours) | 5.2| 5.3 |
>      |AISHELL-1 paper          | 6.44 | 7.62 |
>      | ESPnet2                 |4.4  | 4.7 |
>
> * Noisy versus clean.
>   - The background was only labeled on data of phase 2.
>   - The labeling was conducted in two times (phase 1 and 2) separately. The labeling for phase 1 including subdialect labeling and transcription correction. We didn't think about other info labeling which may distract the auditor from the key task. But for phase 2 which contains only Mandarin, transcription correction was much easier, so we asked the auditor to label the background too (we paid the same money). Labeling on phase 1 again with clean/noisy would be over budget.
>   - In phase 2, clean:noisy equals 92921:292994 (1:3.15), verifying the multi-scenario attribute.
>   - As only half of the whole dataset has that label, we didn't do more deep analysis and left it as auxiliary info for future usage.
>
> * Hosting/maintenance plan.
>   - The downloading method will be published soon after we go through the audit procedure of our company legal dept.
>   - The link https://github.com/KeSpeech/KeSpeech is updated to the paper, and any new info will be published in that repo to make sure the data downloading and other procedures can be easily accessed.
>   - We will host the data at least on our company's website. And OpenSLR is very possible.
>
> * The People’s Speech.
>   - Cited this exciting work in the new version. **(line #30)**
>   - We noticed that work but it's still under review, so we didn't cite it before.

---

### Official Review · Reviewer_Mn5L · 2021-09-17
**A large dataset of speech in Mandarin and dialects**

**Rating:** 8
**Confidence:** 3
**Clarity:** Yes

**Strengths:**

The dataset is large and covers an array of different dialects. The researchers have provided other annotations that may be helpful, including information about each speaker.

**Weaknesses:**

The dataset is not representative and is skewed towards certain groups. For example, the speakers are disproportionately male. It may be the case that downstream applications perform less accurately for groups less well represented in the dataset.

**Additional Feedback:**

N/A

**Correctness:**

The data collection and evaluation procedures appear to be quite thorough. The authors evaluate their dataset using several different comparisons.

**Documentation:**

The authors mention that a download link will be provided but do not provide access to the dataset or any other documentation.

**Ethics:**

It is not clear how volunteers were recruited or incentivized to participate in this study or what protocols were used to get their consent to include their speech in this dataset. I would like to see this issue addressed.

**Relation To Prior Work:**

The authors cite several related works and distinguish their contribution, although I am not familiar enough with this area to make a definitive judgment.

**Summary And Contributions:**

This submission consists of a large dataset containing speech recorded in Mandarin and other subdialects. The researchers collected snippets of text and used volunteers located in different cities in China to read these texts. This dataset will be a useful way to make automated speech recognition systems better able to handle different dialectcs.

---

> ### Author Response · Authors · 2021-09-26
> **Reply to Reviewer Mn5L**
>
> Thanks a lot for your careful review and valuable suggestions.
> We updated the paper and reply to the concerns one by one.
>
>
> * Data imbalance.
>   - There exists some imbalance, e.g., number of female and male, speakers for each city/subdialect and duration for each city/subdialect. That is due to the broker distribution on the Beike platform. **(line #317)**
>
> * Hosting/maintenance plan.
>   - The downloading method will be published soon after we go through the audit procedure of our company legal dept.
>   - The link https://github.com/KeSpeech/KeSpeech is updated to the paper, and any new info will be published in that repo to make sure the data downloading and other procedures can be easily accessed.
>   - We will host the data at least on our company's website. And OpenSLR is very possible.
>
> * How volunteers were recruited or incentivized to participate in this study or what protocols were used to get their consent to include their speech in this dataset.
>   - We recruited volunteers for recording audio from brokers working on the Beike platform. **(line #186)**
>   - We delivered the recoding task to the brokers through a mobile application which was used by all registered brokers on the platform. The application supports different kinds of crowdsourcing tasks including audio recording by reading some text and it records audio with WAV format by 16 kHz sampling rate. **(line #202)**
>   - The participants were rewarded with a simple virtual currency which could only be used on the platform. **(line #206)**
>   - Candidate brokers had to sign an **agreement** for open-sourcing their speech for academic usage before further recording for the two crowdsourcing tasks. **(line #217)**

---

### Official Review · Reviewer_haxc · 2021-09-19
**Exciting work, but has some confusing points and potential problems**

**Rating:** 7
**Confidence:** 4

**Strengths:**

This is a unique dataset that fulfills a niche not covered before. In
particular, beyond focusing on subdialects, the number of speakers is
unprecedented in existing speech-to-text and speaker recognition
corpora.


**Weaknesses:**

Please add a # of hours column to table 2. #Utterances is hard to
interpret on its own.

A serious concern is that there is no comparison of a model trained on
KeSpeech against an existing dataset (AISHELL 1 or 2 is the best
candidate in this case). A common problem with new speech recognition
datasets is that the transcription quality is poor relative to
existing corpora; a good way to verify that this is not the case is,
e.g., to train an ASR model on all dialects of KeSpeech, and then
evaluate it on the AISHELL test set. The word error rate on the AISHELL test set
does not need to be state of the art. It just needs to be reasonable.

Another concern, described below, is that the test sets may not have been created correctly.

**Additional Feedback:**

No additional feedback

**Clarity:**

Phase 2 allegedly occurred two weeks after phase 1, but phase 1 was
stopped when people stopped participating in the task. This
description is quite vague. In particular, someone who volunteered at
the start of phase 1 presumably waited more than two weeks for phase
2.

It is also not clear how many of the same speakers from phase 1 also
appear in phase 2. I understand that it can be hard to clearly
summarize this, but I cannot verify this as a reviewer on my own since
you did not upload the dataset as part of this submission. The number
of speakers who participated in both phases should be straightforward
to compute with a SQL statement.

You claim "The phase 2 task aimed to collect standard Mandarin audio
samples parallel to subdialect ones in phase 1 in terms of some
speaker and some text." It is not clear what "parallel" means in this
case. To create parallel sentences, how did you "translate" text from
a subdialect to standard Mandarin? This seems like a hard thing to do
to me, especially since it would presumably require a human annotator
who knows the subdialect as well as standard Mandarin, but I may be
missing something.


**Correctness:**

I am suspicious of the baseline evaluations in section 4. In
particular, there is no meaningful description of how the test sets
were created. For example, it is good practice to make sure the same
text does not appear in both the train and test sets. In addition, it
is good practice to make sure the same speaker does not appear in both
the train and test sets. There is no description of how test set
creation was done, however.

The results are definitely thorough, very well described, and convincing. It is just worrisome that
a new dataset that could have a lot of future impact may have an incorrectly
prepared test set.

**Documentation:**

Documentation of the dataset format is easy to understand (I speak as
aa long-time kaldi user and developer), except for the
aforementioned vagueness regarding test set creation.

**Ethics:**

I may have a misunderstanding, but it is odd that volunteers were used
for data collection. Normally, datasets created by prompting speakers
to say pre-provided text compensate the speakers (although there are
exceptions to this rule, like for Mozilla CommonVoice, but CommonVoice
is explicitly a non-profit endeavor). I realize that Beike is a good
platform for collecting dialect information given its large geographic
reach, but I cannot help but feel that some information is being
omitted about how brokers using the app were incentivized to
volunteer.

This was said about the sentences collected for volunteers to speak:
"The news genre in CLUECorpus2020-small was crawled from the public We
Media (self-media) platform and it involves almost all common topics
like sports, finance, entertainment and technology." The dataset
itself is licensed under CC-BY-NC-SA. Does that mean that the source
sentences must not have any existing copyright applied to them? I am not sure. It
would be great if an expert could review this.

**Relation To Prior Work:**

Good summary of relation to prior work. Good foundational summary for
someone not already with the subdialects of Mandarin.

**Summary And Contributions:**

The paper introduces a new speech recognition dataset under a
CC-BY-NC-SA license explicitly covering the 8 standard subdialects of
Mandarin.

---

> ### Author Response · Authors · 2021-09-26
> **Reply to Reviewer haxc**
>
> Thanks a lot for your careful review and valuable suggestions.
> We updated the paper and reply to the concerns one by one.
>
> * Add a # of hours column to table 2.
>   - Added.
>
> * Evaluate ASR models trained with KeSpeech on existing test set.
>   - We evaluated the last 3 baseline models (trained on All subdialects, Mandarin and Whole training set respectively) on AISHELL-1 dev/test as shown in Table 3, that verified the reasonability of dataset construction and model building. We updated the paper and also paste the CER% results below for quick reference.
>
>      | Model |  AISHELL-1 dev | AISHELL-1 test |
>      |--|--|--|
>      |All subdialects (ours) | 11.9| 11.9  |
>      |Mandarin (ours)        | 5.7 | 6.0 |
>      |Whole training set (ours) | 5.2| 5.3 |
>      |AISHELL-1 paper          | 6.44 | 7.62 |
>      | ESPnet2                 |4.4  | 4.7 |
>
> * How ASR test set was created.
>   - The development and test sets contain randomly selected 100 and 2,000 speakers with all their audio samples respectively, and the speakers were involved in both phase 1 and 2. **(line #241)**
>   - The train/dev/test sets for Speaker Verification and Subdialect Identification follow the same splitting for ASR with same reorganization. **(line #267,279)**
>   - There's no speaker overlap between training and test sets.
>   - The text of test set has 2220 out of 19723 (about 11%) occurring in training set, but spoken with different speakers and background conditions. We think the ratio is acceptable as the performance without those utts keeps almost the same as before.
>
> * Time interval between Phase 1 and 2.
>   - Two weeks after we stopped the phase 1 task, i.e., no participants could get in and record any more, we started a totally new task (phase 2), so there's at least two weeks between phase 1 and 2.
>
> * The number of speakers who participated in both phases.
>   - There were 17,558 speakers out of total 27,237 (about 64.5%) participating in both phases. **(line #71)**
>   - The overlapped speaker IDs can be easily found with the following files:
>     - Metadata/{phase1.wav.scp,phase2.wav.scp}, contains wav ids for phase 1 and 2 respectively.
>     - Metadata/spk2utt, contains speaker id and their wav ids.
>
> * What "parallel" means.
>   - The phase 2 task aimed to collect standard Mandarin audio samples parallel to subdialect ones in phase 1 in terms of **same** speaker and **same** text. **(line #211)** Sorry for the typo: same -> some.
>   - Parallel = same speaker, same text, different subdialect.
>   - Each subdialect has its own special vocabulary, while vocabulary of standard Chinese constitutes the main part of all subdialects. That is to say, any sentence from standard Chinese can be spoken by Mandarin subdialects. **(line #145)**. So we use text from standard Chinese.
>
> * How volunteers were rewarded.
>   - The participants were rewarded with a simple virtual currency which could only be used on the platform. **(line #206)**
>   - The app is used by almost all registered brokers on the platform.

---

### Comment · Program_Chairs · 2021-10-13
**Official Ethics Review**

It is unclear if this dataset was collected with informed consent - although volunteers signed an agreement for open-sourcing their data. Depending on the open-source licenses being used, such cases can still be ambiguous in the context of machine learning [1]. If possible, it would be great for authors to confirm having gone through some internal institutional review board (IRB) process, or otherwise being deemed exempt.

As this dataset involves sensitive biometric data, we encourage authors to restrict data distribution to prevent malicious use (ie. restricting access by request form, restricted possible use cases for the data, terms of use contract, etc.). Even though anonymized, it is important for authors to monitor how such a dataset is being used and leveraged by other parties and control outsider access to this information.

Although anonymized, malicious actors could still potentially reconstruct certain personal details. It is thus important for authors to include some explicit reflection on the potential risks of the dataset, including the potential for de-anonymization and its use in the development of applications for surveillance.

[1]https://creativecommons.org/2021/03/04/should-cc-licensed-content-be-used-to-train-ai-it-depends/

---

> ### Author Response · Authors · 2021-10-14
> **Reply to Official Ethics Review**
>
>
> Dear Chairs,
>
> Thanks a lot for your careful review and valuable suggestions.
> We deal with the ethical concerns one by one, and also update the paper.
>
> * Informed consent
>   - See the volunteer agreement **(line #216)** and phone interface on
>   https://github.com/KeSpeech/KeSpeech/blob/main/volunteer_agreement.md
>   - The key points involved in the agreement are:
>     - What volunteers need to know before further recording
>       - The audio may include voiceprint information.
>       - Won't store or use any information that can be related or tracked to a specific person.
>       - Press the 'agree' button voluntarily to proceed or EXIT.
>     - What to do for volunteers
>       - Give microphone permission during recording.
>       - Select a city where the volunteer is good at the dialect.
>       - Read the predefined text with Mandarin/Mandarin dialect.
>     - Data usage
>       - Non-commercial.
>       - Academic open-sourcing or challenge.
>       - By Beike internal legally.
>       - By other academic institutions via signing a license, except small sample.
>
> * Internal institutional review
>   - The data collection has gone through the audit of Beike's legal department.
>   - It's strict as:
>     - As a listed company, it disallows any issues, such as privacy leak or illegal text, that may hurt the company's reputation.
>     - National regulation is very strict, with newly enacted Personal Information Protection Law and Data Security Law.
>
> * Biometric info and restricted access/license/use
>   - The biometric info involved in the dataset is only voiceprint. The text was from open-sourced corpus, containing no illegal or sensitive info.
>   - Restricted access/license/use
>     - The company's legal department also suggests a restricted access/license/use and provides a custom license, so we change the CC BY-NC-SA to that one **(line #97)** with https://github.com/KeSpeech/KeSpeech/blob/main/dataset_license.md
>     - The key points are:
>       - Non-commercial. No Distribution.
>       - Compliance. You shall ensure that your use of this dataset does not and will not violate any applicable laws and regulations.
>       - May be updated if necessary.
>   - Usage monitor
>     - For better monitoring the data usage, we hold the dataset on the company's cloud. Any academic institutions who want it will be informed the license and detail their academic usage with a request form, and we will evaluate it carefully.
>     - We will also check if the data is legally used from time to time, by following the work of users or communicating with them directly.
>
> * Explicit reflection on potential risks
>   - We add a subsection 'Potential Risks' to the paper **(line #323)**.
>   - We paste it here for quick reference:
>     - As the text was from open-source corpus with no private or sensitive information, the biggest ethical concern in this dataset is about speaker identity. Although all speaker identity numbers were randomly generated and have no relationship at all with the original ones, there still exists some potential risks that one can de-anonymize that by, for example, comparing the whole dataset to the voice of a specific broker on the platform to retrieve all his/her audio for malicious purposes. On the other hand, all speakers comprise a subset of the Beike brokers, so the platform (if voice authentication is allowed) may be invaded with voice of any one regardless of the specific identity. To reduce the risks as much as possible, we host and maintain the dataset on Beike's own cloud for better monitoring data usage. With the dataset's license, any academic institutions should detail their research plan with a request form, and we will evaluate it carefully. We will also inspect if the data is validly used from time to time, by following the work of users or communicating with them directly.
>
>
> Thanks again for your efforts.

---

### Decision · Program_Chairs · 2021-10-09

**Decision:**

Accept

**Comment:**

This paper introduces a speech recognition dataset for eight standard sub-dialects of Mandarin.

The paper is well-written. The overall feedback from the reviewers were positive. However, reviewers pointed out some of the concerns and weaknesses of the paper. I think the authors addressed some of them well during the rebuttal. I recommend the authors to address them all.

The Reviewer haxc and 91VF have raised some ethical concerns about this paper. The paper collected the speech dataset from the volunteers and certain details about the data collection procedure are not very clear in the paper as pointed out by the reviewers. I would recommend this paper for an ethical review.

Flagged for an additional ethics review because it is not sufficiently clear how volunteers were recruited or incentivized to participate in this study or what protocols were used to get their consent to include their speech in this dataset.